# Lightweight Network DCR-YOLO for Surface Defect Detection on Printed Circuit Boards

**DOI:** 10.3390/s23177310

**Published:** 2023-08-22

**Authors:** Yuanyuan Jiang, Mengnan Cai, Dong Zhang

**Affiliations:** 1School of Electrical and Information Engineering, Anhui University of Science and Technology, Huainan 232000, China; jyyll672@163.com (Y.J.); 2020200688@aust.edu.cn (D.Z.); 2Institute of Environment-Friendly Materials and Occupational Health, Anhui University of Science and Technology, Wuhu 241003, China

**Keywords:** DCR-YOLO, defect detection, printed circuit board, SDDT-FPN, PCR, C_5_ECA

## Abstract

To resolve the problems associated with the small target presented by printed circuit board surface defects and the low detection accuracy of these defects, the printed circuit board surface-defect detection network DCR-YOLO is designed to meet the premise of real-time detection speed and effectively improve the detection accuracy. Firstly, the backbone feature extraction network DCR-backbone, which consists of two CR residual blocks and one common residual block, is used for small-target defect extraction on printed circuit boards. Secondly, the SDDT-FPN feature fusion module is responsible for the fusion of high-level features to low-level features while enhancing feature fusion for the feature fusion layer, where the small-target prediction head YOLO Head-P3 is located, to further enhance the low-level feature representation. The PCR module enhances the feature fusion mechanism between the backbone feature extraction network and the SDDT-FPN feature fusion module at different scales of feature layers. The C_5_ECA module is responsible for adaptive adjustment of feature weights and adaptive attention to the requirements of small-target defect information, further enhancing the adaptive feature extraction capability of the feature fusion module. Finally, three YOLO-Heads are responsible for predicting small-target defects for different scales. Experiments show that the DCR-YOLO network model detection map reaches 98.58%; the model size is 7.73 MB, which meets the lightweight requirement; and the detection speed reaches 103.15 fps, which meets the application requirements for real-time detection of small-target defects.

## 1. Introduction

With the accelerated development of industrialization and intelligent development of electronic devices, the range of printed circuit board applications is expanding. Printed circuit boards contain many kinds of defects [1]. Warpage and shrinkage [2] are common defects of printed circuit boards that occur during the production process, and these two defect types generally lead to variations in the length, width, and thickness of printed circuit boards. Printed circuit boards containing these two kinds of defects are difficult to apply in electronic devices. Common defects in printed circuit boards include surface defects such as missing_hole, mouse_bite, open_circuit, short, spur, and spurious_copper, which may cause damage to electronic devices containing printed circuit boards. They may also lead to major safety incidents. The safety and stability of printed circuit boards have become very important. It is very important to check for defects in printed circuit boards in a timely manner to eliminate potential safety hazards and reduce the number of safety accidents.

Printed circuit board surface defects are small and varied; for the manual visual inspection method, detection efficiency is low and the rate of missed detection is high. For the automatic optical inspection [3] method, the accuracy is low and vulnerable to interference, and the detection speed is slow. With the rapid development of computer technology advances and improved deep learning in the direction of image processing [4,5,6,7,8], researchers have proposed a variety of printed circuit board defect detection methods based on deep learning [9].

One class comprises defect detection methods based on traditional convolutional neural networks [10]. A multiscale feature fusion detection method is implemented based on upsampling and layer-hopping connections [11]. Detection methods are based on basic convolutional neural networks with multiple segmentation of defect regions. These defect detection methods are based on the FasterRcnn multiattention fusion mechanism. The antidisturbance encoder–decoder [12,13,14,15] structure is the basis of the convolutional neural network defect detection methods. Such defect detection methods have complex network structures, a large number of parameters, and a slow detection speed.

The other category is the YOLO [16] family of defect detection methods. Some target detection methods use ResNet as the backbone feature extraction network of YOLOv3 [17]. Other target detection methods are based on the backbone feature extraction network of YOLOv4 [18,19], incorporating long-range attention mechanisms. A defect detection method is based on the YOLOv5 [20,21,22] network with an enhanced perceptual field by introducing a coordinate attention mechanism and enhanced multiscale feature fusion. Based on the improved MobileNetv3 as the backbone feature extraction network, ECAnet [23] is introduced to adjust the feature weights adaptively to enhance the feature extraction of the network. Such defect detection methods have a small number of parameters and a relatively fast detection speed, but for small-target [24,25] defect detection, the feature extraction strength is insufficient and the detection accuracy is low.

Currently, printed circuit board defect detection has two sets of difficulties: (1) Defect detection network models have layers, a complex network model, a large number of parameters, low lightweight [26,27] degree, and slow detection speed. (2) Printed circuit board defect targets are small, and important defect features are difficult to extract, resulting in the low accuracy of small-target defect detection.

In response to the above problems, this paper sets up a double-cross-residual [28] YOLO (DCR-YOLO) defect detection model. It meets the requirements of industrial real-time detection of small-target defects. It also solves the problem of low detection accuracy caused by multiple defect detection network layers, complex model structure, and a low network lightweight degree.

## 2. Approach to the Overall Design of the DCR-YOLO Network Model

Due to the small surface of defects on printed circuit boards, important defect features are difficult to extract. In order to meet the industrial requirements for the accuracy and speed of detection of small-target defects, the detection network needs to be improved for small-target defect detection accuracy and must ensure a high degree of lightness of the detection network. The single-stage YOLO target-detection series of algorithms has good performance, both in terms of detection accuracy and detection speed. The overall structure of the DCR-YOLO network is shown in Figure 1.

The DCR-YOLO network mainly consists of the double-cross-residual backbone (DCR-backbone) module, the pooling-convolution-residual (PCR) module, the same-direction-double-top feature pyramid [29] network (SDDT-FPN) module, the C_5_ECA module, and three prediction heads, namely, YOLO Heads.

### 2.1. Design of the DCR-Backbone Structure

Printed circuit board surface defects are small. In order to be able to fully extract the features of small target defects, enhance the feature extraction ability of small target defects, effectively alleviate the gradient disappearance and explosion problems, and improve the learning ability of the network, a backbone feature extraction network based on Cross-Residual blockbody (CR-blockbody) is designed. The backbone feature extraction network mainly consists of three convolutional structures: two CR-blockbodies and one ordinary Residual-blockbody (R-blockbody) structure. The deeper the backbone feature extraction network, the more feature information of small targets will be lost. The first two backbone feature extraction blocks use CR-blockbody, and the latter one uses R-blockbody, so that the small target feature information is fully extracted and retained, and the model improves the lightness.

The convolution structure is shown in Figure 2a and consists of Conv2D, BN [30] (Batch Normalization), and LeakyRelu activation function. The R-blockbody structure is shown in Figure 2b and consists of four convolution structures, two jump connections, two splicing operations, and one pooling structure. The CR-blockbody structure is shown in Figure 2b and consists of four convolution structures, two jump connections, two splicing operations, and one pooling structure. The CR-blockbody structure shown in Figure 2c consists of six convolutional structures, three jump connections, three splicing operations, and one pooling structure. There are common cross-convolutional layers in the residual block of jump connections, which effectively reduces the problem of feature information loss and depletion when feature extraction is performed on the input feature layer, especially for small target defects.

For the DCR-backbone network module, the first input is a picture feature layer of size 416 × 416 with three channels. After the first convolutional structure, the input image features are transformed from 416 × 416 × 3 to 208 × 208 × 32. After the second convolutional structure, the input features are transformed from 208 × 208 × 32 to 104 × 104 × 64. Next the feature layer is processed by two CR-blockbody structures. The first CR-blockbody structure transforms the input features from 104 × 104 × 64 to 52 × 52 × 128, and the processed feature layer is prepared for YOLO Head-P3 prediction. After the second CR-blockbody structure, the input features are transformed from 52 × 52 × 128 to 26 × 26 × 256 and the processed feature layer is prepared for YOLO Head-P4 prediction. Finally, the feature layer is processed by the R-blockbody structure and the third convolutional structure. The R-blockbody structure and the third convolutional structure transform the input features from 26 × 26 × 256 to 13 × 13 × 512, and the processed feature layer is prepared for YOLO Head-P5 prediction.

### 2.2. Design of the PCR Structure

To enhance the feature fusion mechanism at different scales between the CR-backbone feature extraction network and the SDDT-FPN feature fusion module, and to effectively prevent feature information loss when fusing features at different scales between different modules, the pooled convolutional residual structure CPR was designed. The PCR module mainly consists of a residual convolutional structure and two pooling channel structures. The CBS convolutional structure shown in Figure 3a mainly consists of Conv2D, BN, and Silu activation functions. Both pooling channel structures are composed of two pooling layers of the same size. The pooling layer has a pooling kernel size of 5 × 5, a step size of 1, and a padding number P of 2. The input feature layer is processed by the convolutional structure and the two pooling channels, and then the fused feature layer is passed backwards through a splicing operation. The PCR structure is shown in Figure 3b.

After being processed by the first CR-blockbody structure in the CR-backbone module, the output feature layer is 52 × 52 × 128. The first CBS structure of the PCR module transforms the input feature 52 × 52 × 128 into 52 × 52 × 64. The first pooling channel of the PCR module transforms the input feature 52 × 52 × 64 into 52 × 52 × 64. After the second pooling channel of the PCR module, the input feature 52 × 52 × 64 is transformed into 52 × 52 × 64. After the second CBS structure of the PCR module, the input feature 52 × 52 × 64 is transformed into 52 × 52 × 64. Finally, the processed feature layer is stitched into 52 × 52 × 192 by the splicing operation. Thus, the PCR module’s input feature layer is 52 × 52 × 128, and the output feature layer is 52 × 52 × 192.

### 2.3. Design of the SDDT-FPN Structure

YOLO-Head P5 and YOLO-Head P4 have a deeper backbone feature extraction network, which is suitable for predicting relatively large defects, but the deeper backbone feature extraction network will lead to the loss of some features in the extraction of small target defects, resulting in relatively low accuracy of small target defect prediction. It is more suitable for small-target defect prediction as less features are lost in the process. Therefore, it is necessary to enhance the feature fusion for YOLO-Head P3 to improve the accuracy of small-target defect prediction.

To address the above issues, an isotropic double-top feature pyramid SDDT-FPN structure is designed as shown in Figure 4a. This structure not only facilitates the feature fusion mechanism between the bottom-up special layers, but also re-introduces the isotropic pyramid top for the feature fusion layer where the small-target defect prediction head YOLO Head-P3 is located, further enhancing the feature information transfer in the feature layer for small-target defect prediction. The overall model after the introduction of the SDDT-FPN structure is shown in Figure 4b.

The whole module of DCR-backbone processes the features as 13 × 13 × 512. After the convolution structure on YOLO Head-P5 transforms the input features from 13 × 13 × 512 to 13 × 13 × 256. YOLO Head-P5 to YOLO Head-P4 undergoes a convolution and upsampling operation to transform the feature layer from 13 × 13 × 256 to 26 × 26 × 128. The splicing operation on YOLO Head-P4 transforms the feature layer from 26 × 26 × 128 to 26 × 26 × 384. After the first convolutional structure on YOLO Head-P4, the feature layer is transformed from 26 × 26 × 384 to 26 × 26 × 512. YOLO Head-P4 to YOLO Head-P3, and after the first convolution and upsampling operation, the feature layer is transformed from 26 × 26 × 384 to 52 × 52 × 256. YOLO Head-P4 to YOLO Head-P3. After the second convolution and upsampling operation, the feature layer is transformed from 26 × 26 × 512 to 52 × 52 × 256. After the first splicing operation on YOLO Head-P3, the feature layer is 52 × 52 × 384. After the first convolution structure on YOLO Head-P3, the feature layer is transformed from 52 × 52 × 384 to 52 × 52 × 256. After the second splicing operation on YOLO Head-P3, the feature layer is 52 × 52 × 512.

### 2.4. Design of the C_5_ECA Structure

The overall model has a large amount of feature information to extract and focus on, so to some extent the focus on minor feature information needs to be reduced and the focus on major information needs to be enhanced. The C_5_ECA structure is designed so that the feature extraction and fusion mechanisms are enhanced between the layer structures of the SDDT-FPN network to increase the attention paid to small-target defective feature information. The C_5_ECA module structure is shown in Figure 5a, which mainly consists of two convolutional structures, a residual convolution (consisting of three convolutional structures), an ECAnet [31] structure, and splicing operations.

The first two convolutional structures are mainly used for upsampling operations, and the residual convolutional structure is used for feature extraction and transfer between prediction layer structures to enhance the sensitivity of small-target information extraction. The specific structure of the Effificient Channel Attention network (ECAnet) module is shown in Figure 5b. As can be seen from the ECAnet structure, firstly, the input feature map is globally averaged to pool the h and w dimensions to one, and only the channel dimension is retained. Secondly, a 1D convolution is performed so that the channels in each layer interact with the channels in neighboring layers and share the weights. Finally, using the Sigmoid function for processing, the input feature map is multiplied with the processed feature map weights and the combined weights are assigned to the feature map. After being processed by the ECAnet module, the model is made to adaptively focus on the more important small-target defect feature information, which further improves the adaptive [32] feature extraction capability of the network model and thus improves the prediction accuracy of small-target defects.

The first two convolutional structures in the C_5_ECA module were responsible for upsampling. YOLO Head-P4 to YOLO Head-P3 performed two operations of the C_5_ECA module. For the first operation of the C_5_ECA module, the feature layer of the input module was 26 × 26 × 384, and the feature layer of the output module was 52 × 52 × 256. For the second operation of the C_5_ECA module, the feature layer of the input module was 26 × 26 × 512 and the feature layer of the output module was 52 × 52 × 256.

## 3. Experimental Basis and Procedure

In order to verify the detection performance and prediction performance of the DCR-YOLO model, comparison and ablation experiments were conducted. The experimental data in this paper were obtained from the Human–Computer Interaction Open Laboratory of Beijing University, Beijing, China. This open dataset [33] contains six types of printed circuit board defects required for the experiments. The computer system required for the experiment was Windows 11 operating system and the programming language was Python programming language.

### 3.1. Data Set for the Experiment

This experimental dataset came from the open dataset of printed circuit board defects from the Intelligent Robotics Open Lab, with a total of 10,668 images. First, 2667 images were randomly selected from these experimental data, which contain an equal number of images of six types of printed circuit board surface defects, as shown in Figure 6: missing_hole, mouse_bite, open_circuit, short, spur, spurious_copper. The training set consisted of 2160 images, the validation set consisted of 240 images, and the test set consisted of 267 images.

### 3.2. Evaluation Criteria

The commonly used and representative evaluation metrics used in this paper are Average Precision (*AP*), Mean Average Precision (mAP), Check All Rate (Recall, R), the curve of R change for each defect category, and Frame Rate (Frames Per Second, FPS).

P refers to the proportion of all objects that the model predicts correctly, also known as the accuracy rate, as shown in Equation (1). The check-all rate, R, refers to the proportion of all real objects that are correctly predicted by the model, also known as the recall rate, and is shown in Equation (2). The average precision (*AP*) formula is shown in Equation (3) and refers to the size of the area enclosed by the two curves, the precision P curve and the accuracy R curve, on the interval (0, 1). The mean accuracy (mAP), which is the average of the *AP* values for all categories, reflects the overall effectiveness and overall accuracy of the model, and is an important overall measure of the model’s performance.
(1)$$\mathrm{Pre}\;cision=\frac{TP}{TP+FP}$$
(2)Recall=TPTP+FN
(3)$$AP=\int_{0}^{1}P(R)d(R)$$

The formula *TP* indicates that the sample is a positive sample, representing the number of samples predicted to be positive; the *FP* value represents a negative sample, indicating the number of samples predicted to be positive; *FN* means that the sample is a positive sample, and the number of samples predicted to be negative is wrong.

### 3.3. Experimental Platform and Parameters

The configuration and parameters required for the experiments are as follows: the framework for deep learning is Pytorch 1.12.1 + CUDA116. Pytorch is a deep learning framework developed at Facebook’s Artificial Intelligence Research Lab in Menlo Park, San Mateo County, CA, USA. CUDA is a parallel computing platform and programming model from NVIDIA in Santa Clara, CA, USA. This study used Python version 3.8, which is a computer programming language invented by Guido van Rossum of the Netherlands Research Society for Mathematics and Computer Science. The operating system is Windows 11, which is a computer operating system invented by the Microsoft Corporation in Redmond, WA, USA. The graphics processor is an NVIDIA GeForce RTX 3050Ti GPU with 4G video memory, and the relevant parameters for training are shown in Table 1.

### 3.4. Model Training Process and Results

The DCR-YOLO model training process loss change curve and training process map change curve are shown in Figure 7a. After 350 epochs of training, the loss curve shows that the loss function is basically in a converged state after 200 epochs, while the map change curve shows that the average accuracy value of the model reaches about 90% at 175 epochs, and the map value is basically smooth after 260 epochs. The average accuracy curves of each category after training are shown in Figure 7b, and the average accuracy of each category is basically above 95%.

### 3.5. Ablation Experiments with Different Modules

Ablation experiments were designed to evaluate the degree of optimization of the performance of the algorithm with different combinations of designed modules. The results of the ablation experiments are shown in Table 2. DCR-3Head indicates a base model consisting of a CR-backbone network and three predictor heads. Px-PCR indicates the introduction of a PCR module after the backbone feature layer in which a predictor head is located. sddt-fpn indicates the introduction of an sddt-fpn module between the backbone feature extraction network and the predictor head. Meanwhile, 1-C_5_ECA indicates the introduction of a C_5_ECA module between P5 and P4. 2-C_5_ECA indicates the first (left) introduction of the C_5_ECA module between P4 and P3, while 3-C_5_ECA indicates the second (right) introduction of the C_5_ECA module between P4 and P3.

Experiments 1–9 combine different modules. Experiment 1: DCR-backbone + 3-YOLO Head, Experiment 2: DCR-backbone + SDDT-FPN + 3-YOLO Head, Experiment 3: DCR-backbone + SDDT-FPN + P_3_-PCR + 3-YOLO Head, Experiment 4: DCR-backbone + SDDT-FPN + P_4_-PCR+3-YOLO Head, Experiment 5: DCR-backbone + SDDT-FPN + P_5_-PCR + 3-YOLO Head, Experiment 6: DCR-backbone + SDDT-FPN + P_3_-PCR + 1-C_5_ECA + 2-C_5_ECA + 3-YOLO Head, Experiment 7: DCR-backbone + SDDT-FPN + P_3_-PCR + 1-C_5_ECA + 3-C_5_ECA + 3-YOLO Head. Experiment 8: DCR-backbone + SDDT-FPN + P3-PCR + 2-C_5_ECA + 3-C_5_ECA + 3-YOLO Head, Experiment 9: DCR-backbone + SDDT-FPN + P_3_-PCR + 1-C_5_ECA + 2-C_5_ECA + 3-C_5_ECA + 3-YOLO Head. The specific results of the experiment are shown in Table 2.

From Experiment 1, it can be seen that the network structure of the CR-backbone network-based model map = 96.76, proving that the designed cross-residual CR-blockbody has a strong capability for small-target defect feature extraction.

As can be seen from Experiments 1 and 2, the introduction of the SDDT-FPN module improved map and R by 0.5% and 1.37%, respectively, compared to Experiment 1, proving that the SDDT-FPN enhanced the feature fusion mechanism between layers and also improved the feature fusion capability for the feature fusion layer in which the small-target prediction head YOLO Head-P3 is located, further improving the small-target defect detection accuracy.

From Experiments 3, 4 and 5, it can be seen that the introduction of the PCR module after the backbone feature extraction layer in which the P3 predictor head is located gives the best results in terms of fusing the different scale feature layers, with a 0.74% improvement in MAP, compared to Experiment 2.

From Experiments 6, 7, 8 and 9, it can be seen that the introduction of two (left and right sides) C_5_ECA models between predictor heads P4 and P3 in the SDDT-FPN structure strengthens the feature fusion mechanism between the layer structures of the SDDT-FPN network and improves the attention to small-target defect feature information, with map and R improving by 0.58% and 0.48%, respectively, compared to Experiment 3.

### 3.6. Comparative Experiments with Different Models

To verify the model DCR-YOLO feasibility and effectiveness, six current mainstream target detection models were trained and tested on the printed circuit board defect dataset using YOLOv3, YOLOv4, YOLOv4-tiny, YOLOv5-s, YOLOv5-m, and YOLOv7-tiny [34] in the same experimental environment, and the results are shown in Table 3.

As can be seen from Table 3, for model detection in terms of MAP, the DCR-YOLO model reached 98.58%, which was 10.14%, 1.44%, 8.95%, 4.24%, 2.69%, and 3.26% higher than YOLOv3, YOLOv4, YOLOv4-tiny, YOLOv5-s, YOLOv5-m, and YOLOv7-tiny, respectively, 3.26%. The DCR-YOLO model R = 97.24%, which is 25.05% higher than YOLOv5-s and 5.37% higher than YOLOv4; this is a significant improvement compared to the other models. The volume of the model is 7.73 MB, which is 56.23 MB and 13.34 MB lower than the YOLOv4 and YOLOv5 m models, respectively, meeting the lightweight requirement. The inspection speed of fps = 103.15 meets the demand for real-time inspection of printed circuit board defects.

Overall, the comparison of the experimental results with several current mainstream target detection models highlights the superiority and improvement of the DCR-YOLO model. Table 3 shows that the best current target detection model is YOLOv4, which has a map value of 97.14. This is 1.41 percentage points lower than the map value of the DCR-YOLO model. In terms of R-value, the best model, YOLOv4, is 5.37 percentage points lower than the DCR-YOLO model. In terms of model size, the DCR-YOLO model is 56.23 MB smaller than the best model, YOLOv4. In terms of detection speed, the DCR-YOLO model can detect 71.79 frames more images per second than the best model, YOLOv4. The DCR-YOLO model is superior to several current target detection models in terms of map and R, and its speed of detection is greatly improved compared to current models.

## 4. Visualization of Results Analysis

The DCR-YOLO model was conceived based on the structure of the YOLO series of models. In order to verify the actual detection effect of the DCR-YOLO model, six types of printed circuit defect images were randomly selected for detection and also compared with the detection results of the more lightweight YOLOv4-tiny model. The detection results of the YOLOv4-tiny model are shown in Figure 8 and the detection results of the DCR-YOLO algorithm model are shown in Figure 9.

The comparison of the two models shows that for (a) missing_hole and (b) mouse_bite, both the original model and DCR-YOLO detected the defects that were present, and there were no missed detections, and the DCR-YOLO model and the YOLOv4-tiny model were equal in terms of detection accuracy. For (c) open_circuit, (d) short, (e) spur, and (f) spurious_copper, the YOLOv4-tiny model showed no leakage, while the DCR-YOLO model showed no leakage, and the average accuracy of the detected defects was higher than that of the YOLOv4-tiny model. The DCR-YOLO model is superior to the YOLOv4-tiny model.

## 5. Conclusions

In this paper, the DCR-YOLO model is designed. It solves the problem of difficult feature extraction due to small defect targets on printed circuit boards and improves the accuracy of surface-defect detection on printed circuit boards. The overall structure of the inspection model is simple and lightweight, while also meeting the real-time inspection speed requirements.

The experimental results show that the most basic network structure consisting of a CR-backbone backbone feature extraction network and three YOLO-Heads achieves a Map of 96.76% and a recall of R of 95.38%, indicating that the designed cross-residual CR-blockbody has a strong feature extraction capability for small-target defects. The PCR module effectively bridges and fuses the feature maps of different sizes between the DCR-backbone feature extraction network and the SDDT-FPN structure. The C_5_ECA module focuses on the small-target defect information of the printed circuit board, further enhancing the feature fusion and transfer capability between the SDDT-FPN structure layers, and improving the adaptive feature extraction capability of the network model, enhancing the convergence capability of the network to a certain extent and improving the prediction accuracy. 

The DCR-YOLO model has significant advantages over several current mainstream target detection models in terms of detection accuracy, recall, model size, and detection speed. Compared to YOLOv4, which is the most effective of several current target detection models, the DCR-YOLO model provides a 1.41% improvement in terms of map value and a 5.37% improvement in terms of model recall R. The DCR-YOLO model reduces the size of the model by 56 MB in terms of detection speed compared to YOLOv4. The DCR-YOLO model improves the recall of the model by 5.37% and reduces the size of the model by 56.23 MB. In terms of detection speed, the DCR-YOLO model detects 71.79 frames per second more than the YOLOv4 model, making it feasible to detect small defects in real time.

The current model needs to be further improved in terms of lightness so that it can be more easily embedded in more mobile terminals. The DCR-YOLO model aims to detect surface defects in printed circuit boards. The combination of internal performance design analysis methods and detection methods for surface defects of printed circuit boards is yet to be investigated. For the improvement of comprehensive performance improvement of printed circuit boards, the combination of internal performance analysis method of pattern recognition [35] and surface-defect detection method of DCR-YOLO model offers a better outlook.

## Figures and Tables

**Figure 1 sensors-23-07310-f001:**
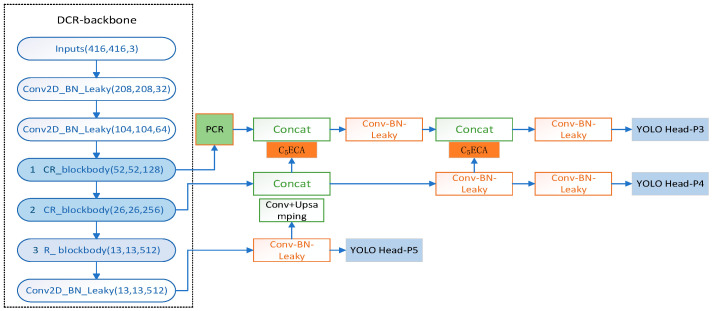
DCR-YOLO network structure.

**Figure 2 sensors-23-07310-f002:**
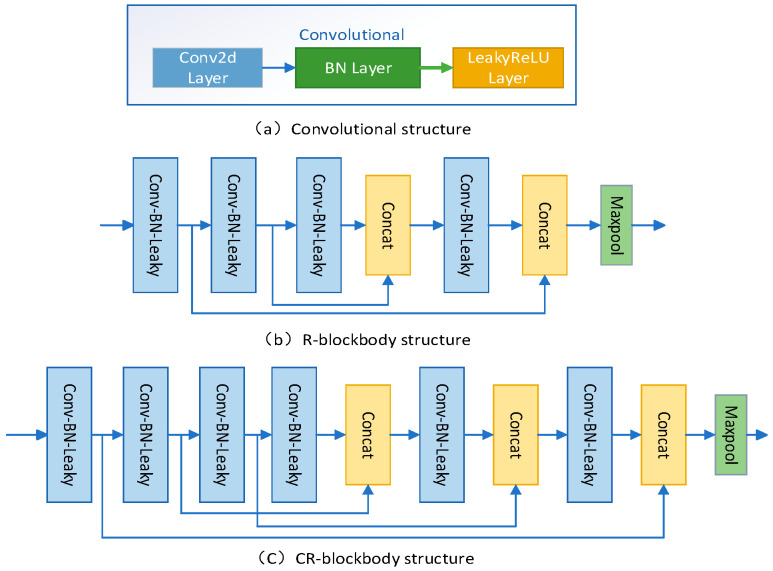
DCR-backbone structure.

**Figure 3 sensors-23-07310-f003:**
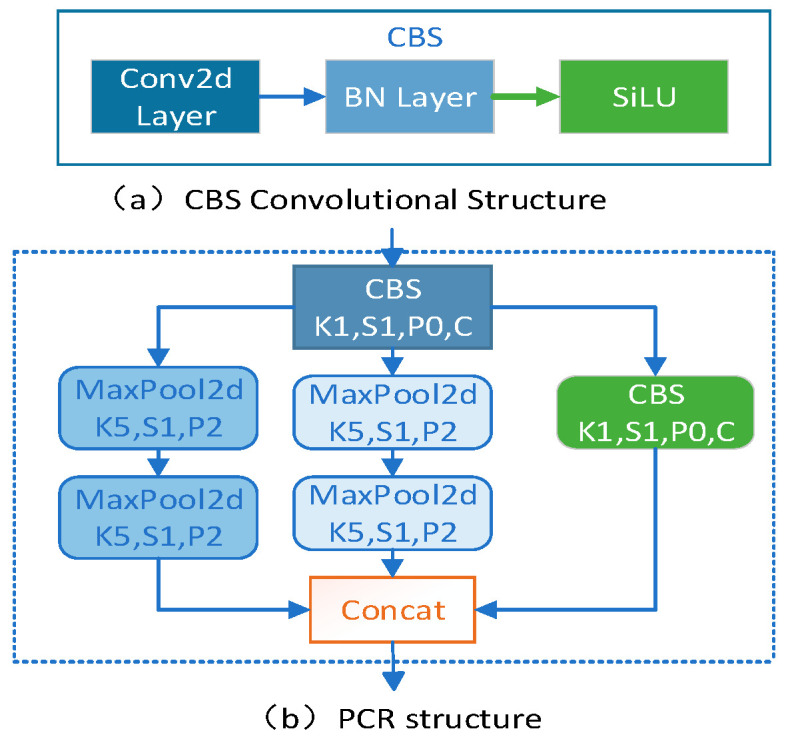
PCR structure.

**Figure 4 sensors-23-07310-f004:**
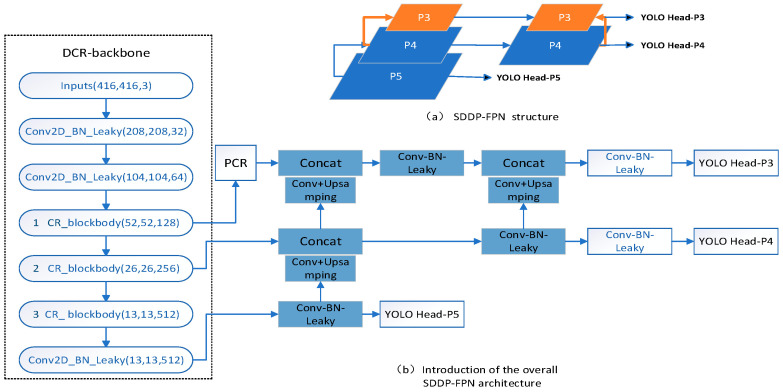
SDDT-FPN structure.

**Figure 5 sensors-23-07310-f005:**
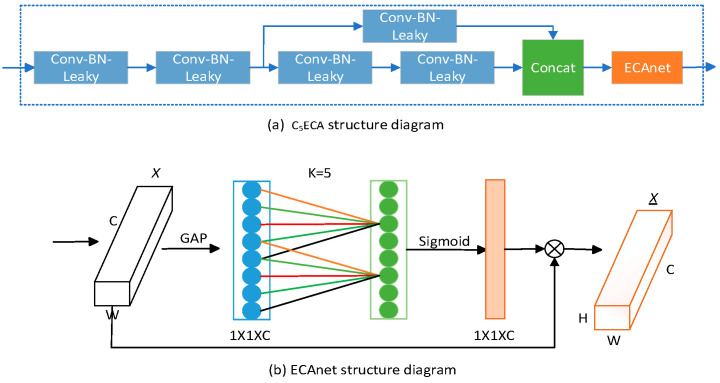
C_5_ECA structure.

**Figure 6 sensors-23-07310-f006:**
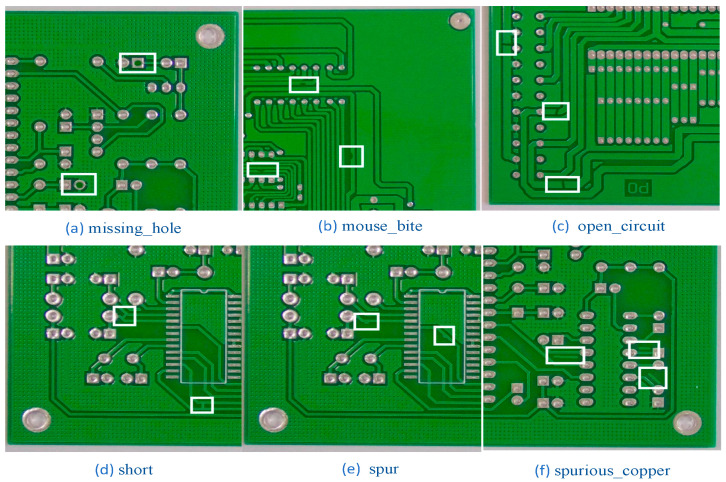
Defect diagram for 6 types of printed circuit boards. (The white boxes contain defects on the surface of the printed circuit board).

**Figure 7 sensors-23-07310-f007:**
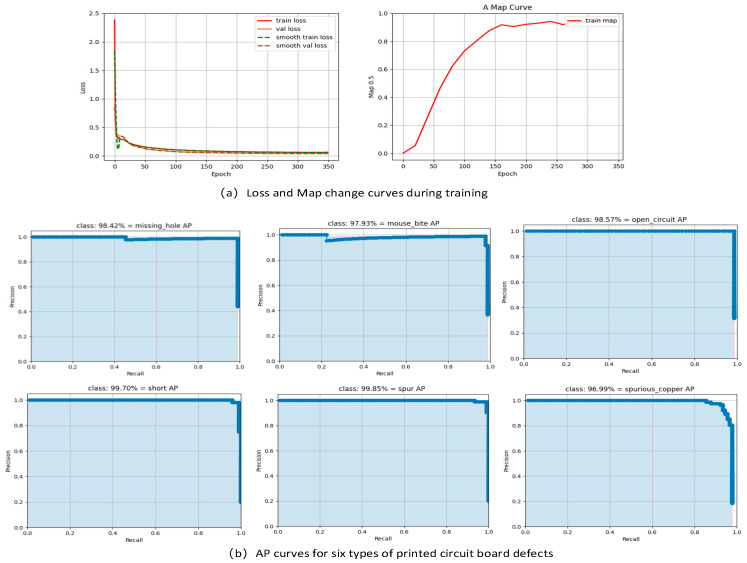
The process and results of the training.

**Figure 8 sensors-23-07310-f008:**
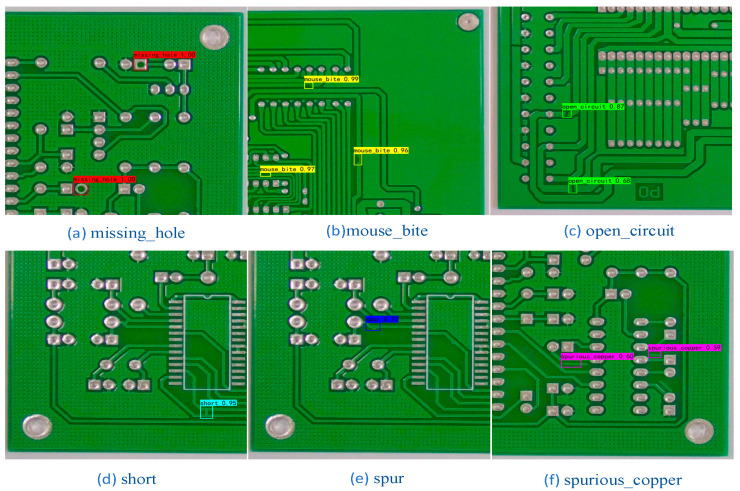
YOLOv4-tiny model test results.

**Figure 9 sensors-23-07310-f009:**
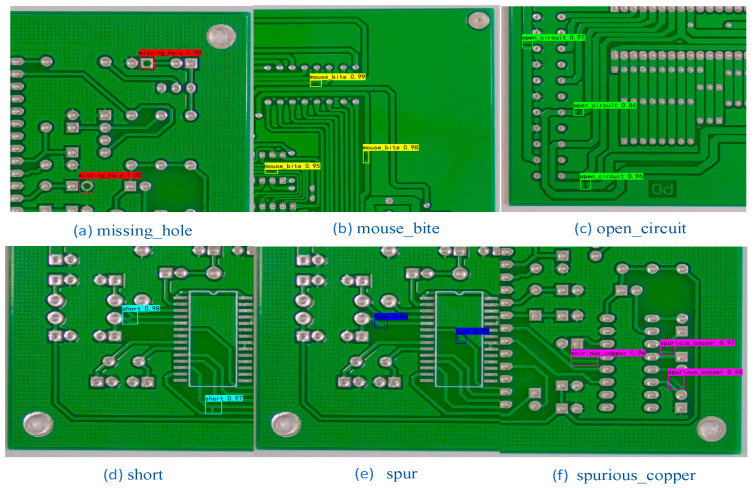
DCR-YOLO model test results.

**Table 1 sensors-23-07310-t001:** Training-related parameters.

Parameters	Numerical Values
Original image size	604 × 604
Training size	416 × 416
Initial learning rate	0.01
Batch size	4
Optimizer type	SGD Optimizer

**Table 2 sensors-23-07310-t002:** Results of ablation experiment.

Number	DCR-3Head	SDDT-FPN	P_3_-PCR	P_4_-PCR	P_5_-PCR	1-C_5_ECA	2-C_5_ECA	3-C_5_ECA	Map/%	R/%	FPS
1	√								96.76	95.38	123.47
2	√	√							97.26	96.75	117.08
3	√	√	√						98.00	96.76	112.55
4	√	√		√					97.54	96.80	112.74
5	√	√			√				97.87	96.67	113.46
6	√	√	√			√	√		98.16	97.35	105.17
7	√	√	√			√		√	98.47	97.13	106.00
8	√	√	√				√	√	98.58	97.24	103.15
9	√	√	√			√	√	√	97.83	96.30	102.66

**Table 3 sensors-23-07310-t003:** Comparison of experimental results.

Model Name	Map/%	R/%	Model Volume/MB	FPS
YOLOv3	88.44	66.33	61.55	39.35
YOLOv4	97.14	91.87	63.96	31.36
YOLOv4-tiny	89.63	79.95	5.89	170.43
YOLOv5-s	94.34	72.19	7.08	72.69
YOLOv5-m	95.89	79.86	21.07	38.98
DCR-YOLO	98.58	97.24	7.73	103.15
YOLOv7-tiny	95.32	80.33	6.03	98.61

## Data Availability

The data presented in this study are available on request from the corresponding author.

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
