# Peer review of "Lightweight Network DCR-YOLO for Surface Defect Detection on Printed Circuit Boards"

_sensors, 2023, doi:10.3390/s23177310_

Round 1
Reviewer 1 Report
The manuscript solves a very interestin problem. The authors clearly described the proposed algorithm network structure and provided a reasonable explanation of the role of the new structure.
But there is an important issue that the author needs to clarify. That is is DCR or CR used in the proposed network?
Moreover, the Section 5 shoulde be Conclusion, not be Discussion.
Author Response
Please refer to the annex

Reviewer 2 Report
The authors in the paper present a DCR-YOLO surface defect detection network for printed circuit boards. The network mainly consists of the following modules: Double Cross Residual backbone module, Pooling Convolution Residual module, Same Direction Double Top Feature Piramyd Network module, C5ECA module, and three YOLO head prediction heads. The proposed network improves the accuracy of surface defect detection and proposes to solve the problem of feature extraction due to the small size of defects appearing on printed circuit boards. Validation for the proposed network was done using 2667 images from a publicly available printed circuit board defect dataset from the Open Laboratory for Intelligent Robotics. The images contained the following defects: missing hole, mouse bite, open circuit, shorts, spur and, spurious copper.
It is an interesting approach to a real problem, that of real-time detection of surface defects on printed circuit boards. The presentation of the proposed system, even if it does not go into much detail, is clear and provides sufficient information about the proposed network.
I would still have a few observations/questions related to the content:
- in the paragraph preceding equations 1, 2, and 3 reference is made to the parameters P (accuracy rate), and R (recall rate) but the terms "Precision" and "Recall" appear in the equations; are they the same?
- in section 3.5, line 257 refers to Table 2 but the experiments appear to be presented in Table 3;
- in the next section (3.6) reference is made to mainstream models used for comparison and the data are presented in Table 3, but they appear in Table 2;
- are there anywhere defined in the specialized literature the limits of the parameters - volume and fps - that allow an objective framing of networks as lightweight and ready for real-time detection or is the placement in these categories based on a comparison between the parameter values in Table 2?
The paper needs moderate language editing. It's tough to follow in its current form. A few problems will be mentioned below, but the list is not complete:
- additional text at line 291, "and the experimental results are shown in Table The experiment results are shown in Table 3";
- certain sentences start with a lowercase letter, such as lines 20, 105, 107, 109, .....
- Figure 9 caption is missing a D from DRC
Overall, maybe It might be a good idea to have the text reviewed by someone who has better knowledge of the English language to to improve the quality of work.
Author Response
Please refer to the annex

Reviewer 3 Report
Dear Authors, let me compose my criticism for the paper: Lightweight network DCR-YOLO for surface defect detection on printed circuit boards.
The paper is nice, it contains valuable information, but i suggest however minor points for improvement:
C1 - the introduction lacks a thorough discussion and literature survey for surface defects. I also lack the surface defect of warpage and PCB shrinkage, which is also an interest of network-based investigations. (e.g. 10.1108/SSMT-07-2014-0014)
C2 - avoid pushed figures like Fig 2, where the presentation shows a wrong picture ratio compared to the originally composed picture. Later this is apparent in more figures.
C3 - please highlight any used tool's origins (e.g. (OLIR [32,33] - > country/company/version is it coming from).
Minor spellcheck is required only, flow is OK!
Author Response
Please refer to the annex

Reviewer 4 Report
The attention of the authors for improving Printed Board circuits is appreciated. The introducto is well posed. The procedure is clearly discussed.
THe results are well descided.
Moreover the following items are suggeste to improve tha paper:
1) The authors must explain clearly the improvements respect the existing methods.
2) Moreover the techniques could be joined with other strategiews well conceived in the PB behavior and not only to the control,
I refer also to the design techniques. Therefore it is useful to remark this aspect and to include the following regerence:
IEEE AccessOpen AccessVolume 10, Pages 31760 - 317742022 Document type Article• Gold Open Access Source type Journal ISSN 21693536 DOI 10.1109/ACCESS.2022.3160449Model Identification to Validate Printed Circuit Boards for Power Applications: A New Technique
- Bucolo, Maidea, b;
- Buscarino, Arturoa, b;
- Famoso, Carloa;
- Fortuna, Luigia, b;
- Frasca, Mattiaa, b;
- Cucuccio, Antonioc;
- Rascona, Gaetanoc;
- Vinci, Giovannic
Author Response
Please refer to the annex

Reviewer 5 Report
This manuscript describes the implementation of DCR-YOLO model for feature extraction and thus improve the defect detection accuracy of PCB board. Authors have gone throught the model with details and showed relative experiment results. The results presented in this manuscript are interesting while there are some revisions needed to be addressed:
1. In section 2, how the input features are determined for each structure? Please explain with more details on the model initialization.
2. Recommend to add grids for fig.7(b) in order to see more precise values.
3. What are the experiments 1-9 mentioned in section 3.5? Please describe the DOEs.
4. Authors mentioned current detection methods are with low detection accuracy and slow speed. Please provide a comparison table to compare the results in manuscript vs. current best techniques.
1. Line 20, please capitalize "t" in "the C5ECA" as it is the start of a sentence.
2. Line 39, remove 2 "is" after "detection efficiency" and "missed detection".
3. Line 40, add "and" before "detection speed is slow".
4. Line 47-48, "Detection methods based ..." misses verb.
5. Line 48-49, "Defect detection method based ..." misses verb.
6. Line 49-51, "Anti-disturbance encoder, decoder structure of convolutional..." misses verb.
7. Line 53-54, replace "using" with "use".
8. Line 56-58, sentence misses verb.
9. Line 93, replace "," with "." before "in order to ...". Otherwise, the whole sentence from ln93 to ln97 contains two verbs.
10. Line 105, capitalize "t" for the start of sentence "the R-blockbody .."
11. Line 126, capitalize "t" for the start of sentence "the CPR module .."
12. Line 127, capitalize "t" for the start of sentence "the CBS convolutional .."
13. Line 164, capitalize "t" for the start of sentence "the C5ECA module .."
14. Line 170, capitalize "t" for the start of sentence "the specific structure of .."
Author Response
Please refer to the annex

Round 2
Reviewer 1 Report
The Chinese in Figure 3 needs to be changed to English.
Author Response
Please refer to the annex
